# How Women Perceive Severity of Complications after Pelvic Floor Repair?

**DOI:** 10.3390/jcm11133796

**Published:** 2022-06-30

**Authors:** Anne-Cécile Pizzoferrato, Stéphanie Ragot, Louis Vérité, Nicolas Naiditch, Xavier Fritel

**Affiliations:** 1Department of Obstetrics and Gynaecology, Caen University Hospital Center, 14000 Caen, France; 2INSERM CIC 1402, Poitiers University, 86021 Poitiers, France; stephanie.ragot@univ-poitiers.fr (S.R.); louis.verite01@gmail.com (L.V.); xavier.fritel@univ-poitiers.fr (X.F.); 3Predictive Research in Spine/Neuromodulation Management and Thoracic Innovation/Cardiac Surgery Laboratory (PRISMATICS), Poitiers University Hospital, 86021 Poitiers, France; nicolas.naiditch@gmail.com; 4Department of Obstetrics and Gynaecology, La Miletrie University Hospital, 86000 Poitiers, France

**Keywords:** urinary incontinence, pelvic organ prolapse, surgery, complications

## Abstract

Background: The Clavien-Dindo classification, used to describe postoperative complications, does not take into account patient perception of severity. Our main objective was to assess women’s perception of postoperative pelvic floor repair complications and compare it to the classification of Clavien-Dindo. Methods: Women and surgeons participating in the VIGI-MESH registry concerning pelvic floor repair surgery were invited to quote their perception of complication severity through a survey based on 30 clinical vignettes. For each vignette, four grades of severity were proposed: “not serious”, “a little serious”, “serious”, “very serious”. Results: Among the 1146 registered women, we received 529 responses (46.2%) and 70 of the 141 surgeons (49.6%) returned a completed questionnaire. A total of 25 of the 30 vignettes were considered classifiable according to the Clavien-Dindo classification. The women’s classification was concordant with Clavien-Dindo for 52.0% (13/25) of the classifiable vignettes. The women’s and surgeons’ responses were discordant for 20 of the 30 clinical vignettes (66.7%). Loss of autonomy (self-catheterization, long-term medication use) or occurrence of sequelae (organ damage or severe persistent pain) were perceived by women as more serious than Clavien-Dindo classification or than surgeons’ perceptions. Conclusions: Women’s perception of pelvic floor repair surgery seems different from the Clavien-Dindo classification. Lack of repair and long-term disability seem to be two major factors in favor of perception of the surgical complication as serious.

## 1. Introduction

Pelvic floor disorders (PFD), including urinary incontinence (UI) and pelvic organ prolapse (POP), are common and may impact a woman’s quality of life because of physical, psychological, social, and sexual repercussions [1,2,3].

Although surgery is currently one of the main treatment options, postoperative complications may be serious and difficult to manage, particularly in the case of mesh complications such as mesh exposure, dyspareunia, or chronic pelvic pain. In this context, the VIGI-MESH registry was created in France in 2017 to capture complications following pelvic floor repair with or without mesh placement [4]. 

The most widely used classification to describe postoperative complications is the Clavien-Dindo classification [5,6]. It is based on the degree of invasiveness of the therapy required to treat a complication and is simple to use, adaptable to different specialties, and standardized to avoid subjectivity. In 2018, the Clavien-Dindo classification was validated by the European Association of Urology (EAU) for the grading and reporting of postoperative complications in urology after an online survey of urologic scenarios found moderate to high interrater agreement among their members [7]. To complete this classification, the EAU published a classification of intra-operative events in 2020 [8].

However, these classifications do not take into account patient perception of severity, while some studies have linked patient dissatisfaction to a patient’s feeling of being “unprepared for surgery” [9,10]. 

Therefore, an approach based on the patient’s perception of the severity of perioperative complications would be informative for the surgeon and could improve patient satisfaction. Our main objective was to assess women’s perception of surgery complications after surgery for UI or POP and compare it to the Clavien-Dindo classification. 

## 2. Materials and Methods

We invited the women included in the VIGI-MESH registry to participate in a clinical vignette-based survey to assess their perception of complication severity. VIGI-MESH was set up in several French surgical centers to follow the surgical activity of UI and POP (with or without mesh) and to assess serious complications and surgical management for recurrence [4]. Complications were defined in the registry according to the Clavien-Dindo classification, which includes five main grades [6]:

Grade I: variations from normal postoperative follow-up including analgesics and bladder catheterization;

Grade II: including pharmacological treatment with drugs other than those allowed for grade I complications—anticoagulants, antibiotics, local anesthesia, and blood transfusions;

Grade III: surgical, endoscopic, or radiological procedures;

Grade IV: requiring intensive care unit care;

Grade V: death.

The survey consisted of 30 clinical vignettes presented in a random order and based on the complications described in the registry. Two reminders were sent out if the women did not respond to the first request. 

For each vignette, four grades of severity were proposed: “not serious”, “a little serious”, “serious”, and “very serious”. The women could explain their answers with a free commentary. The same questionnaire was completed by surgeons participating in the registry. 

A pretest was performed in a sample of 60 patients to estimate the understanding of the clinical vignettes and modifications were made according to the women’s comments.

To compare their responses with the Clavien-Dindo classification, two of the authors (XF and ACP) rated the vignettes using the Clavien-Dindo grades: some vignettes were easily classifiable, whilst others could be classified by analogy to situations described by Clavien-Dindo. When we had no resources, the clinical vignette was considered to be unclassifiable.

For this study, we conducted a descriptive and qualitative analysis: the responses to the questionnaire were described as relative frequencies and plotted in a histogram. We considered that the responses of the women or the surgeons were concordant with the Clavien-Dindo classification when the majority of them answered: “not serious” for a grade I complication, “a little serious” for a grade II complication, “serious” for a grade III complication, and “very serious” for a grade IV complication. We completed the comparison between women and surgeons’ responses with a Fisher’s exact test. 

A qualitative analysis of the women’s comments was performed, and the comments were classified into four domains: (1) dependency or constraint, (2) humiliation or distress, (3) disappointment or failure, (4) worry or danger. 

## 3. Results

The questionnaire was sent by email link in January 2021 to the 1146 women whose email address was provided at inclusion in VIGI-MESH. The majority of women had their first surgery at the time of inclusion in the VIGI-MESH registry, which started in 2017 (96.4%). Forty-one women had their initial surgery before 2017 (27 between 2010 and 2016; 14 before 2010). 

After the two reminders, we received 529 completed questionnaires (46.2%). The mean age of the respondents was 62.4 (±11.7) years. A total of 222 of the respondents (42.7%) had mid-urethral sling (MUS) placement alone, 246 (47.4%) had POP repair alone, and 51 (9.8%) had POP repair and MUS. POP repair was vaginal native tissue repair for 14.8% (44/297) of women, transvaginal mesh for 31.0% (92/297), and abdominal mesh for 54.2% (161/297). 

Of the 529 respondents, 173 (32.7%) did not have the baccalauréat (French certificate on completion of high school), 237 (44.8%) had the baccalauréat, a Bachelor’s degree or equivalent, and 119 (22.5%) had a Master’s degree or higher educational level. There was no significant difference between respondents and non-respondents in terms of body mass index, menopausal status, type of surgery, or history of complications (declared in the registry by surgeons). Respondents were significantly younger than non-respondents (Table 1). 

A total of 80 of the 529 respondent women (15.1%) reported having experienced a surgical complication. There was no significant difference in the responses to the vignettes between the women who had a complication or not.

Of the 141 surgeons who were also contacted by email, 70 (49.6%) returned a completed questionnaire.

A total of 25 of the 30 vignettes were considered classifiable according to the Clavien-Dindo classification (4 grade I vignettes, 5 grade II, 14 grade III and 2 grade IV). The remaining five were considered unclassifiable because they reported perioperative complications, prolongation of hospitalization, or termination of the planned surgery because of technical difficulties. Eleven vignettes reported complications related to mesh placement. 

The women’s and surgeons’ classifications were concordant with Clavien-Dindo for 52.0% (13/25) and 60.0% (15/25) of the classifiable vignettes, respectively. 

Among the Clavien-Dindo grade I complications (Table 2), women considered catheter placement for a few days (vignettes #13 or 24) as not serious or slightly serious, and self-catherization (#4) as serious; additionally, 50.8% of the women who added a comment (*n* = 313) about vignette #4 mentioned that this situation represented a “constraint” or a “dependency”, and 12.1% as “humiliating” or “causing mental distress”. Among the grade II complications, women perceived the long-term use of anticoagulation for pulmonary embolism (vignette #6) as very serious, and stiches placement under local anesthesia (vignette #3) as not serious. The women considered the vignettes including severe postoperative pain (#1 and 16) as serious, and persistent severe vaginal pain after complication management (# 20) as very serious. The comments about vignette #20 included a feeling of “dependency” or “constraint” (32.4%), “disappointment” or “failure” (26.7%), a feeling of “danger” or “worry” (26.3%), and “humiliation” or “prejudice” (19.0%). The comments of the three vignettes describing removal of the sling (#1, 10, and 28) showed that the women were worried about the risk of SUI recurrence and considered it as serious.

For the vignettes describing perioperative events (unclassifiable) (Table 3), women reported that it was not serious if the planned surgery had been carried out, but serious if the planned surgery had not been performed.

Using the most answered item, women’s and surgeon’s responses were discordant for 10 of the 30 clinical vignettes (33.3%). Using the Fisher’s exact test, the distribution of responses was significantly different between women and surgeons for 20 vignettes (66.6%). Six vignettes (20.0%) were considered less serious by the women than the surgeons including complete mesh removal (#17,27,29) and prolonged hospitalization or early reintervention (#18,23,26). Fourteen vignettes (43.3%) were considered more serious by the women including scar troubles (#3, 5, 19), prolonged disorders (#4, 6, 9, 12), POP or UI recurrence (#16, 28) and termination of a laparoscopic procedure because of adhesions (#7, 25 and 30) (Table 2). 

## 4. Discussion

Our study suggests that women’s perception of complications following pelvic floor repair surgery is not concordant with the Clavien-Dindo classification. Furthermore, women do not perceive perioperative events as being serious if the surgery goes ahead as planned.

Our results indicate that the duration and burden of complication management before recovery is associated with a more serious perception. For example, women in our study perceived anticoagulation for pulmonary embolism to be very serious; and surgeons also classified this complication as more serious than the Clavien-Dindo classification. Women considered that placement of a urinary catheter for a few days was less serious than self-catheterization for chronic retention; daily self-catheterization was perceived by some as a condition that impacts their self-image (“humiliating”, “causing mental distress”). The study by Corbussen-Boekhorst et al. found similar comments in a qualitative semiconstructed interview study conducted to assess barriers for patients dealing with self-catheterization in everyday life [11]. However, compared with self-catheterization, women perceive the placement of a JJ ureteral stent as being a less serious complication; perhaps because there is no significant impact on their everyday life.

The degree of pain, and particularly dyspareunia that can impact a woman’s sexual quality of life, emerged as a strong determinant for classifying the severity of the complication. This confirms the work by Rendell et al. who assessed the concordance between patient perception and a complication scale, and found that the level of pain was a determining factor of patient perception of complication severity [12].

In the three vignettes describing cases of sling removal with the same Clavien-Dindo grade (grade III), women considered the complication as more serious if there was a risk of SUI recurrence. Thus, the long-term uncertainty concerning the risk of recurrence of symptoms after management of the complication seems more important than the notion of surgical revision.

Our results showed that women did not consider an intraoperative complication as serious if the surgery went ahead, even if it was not according to the planned technique. Conversely, if the planned surgery was not performed, even in the absence of complications, the women considered it as serious. In another qualitative study, women interviewed preoperatively expressed the need to “get back to normal” within the time period specified by their surgeon [13]; in the postoperative group, women categorized symptoms such as incontinence or constipation, or lack of improvement in sexual function as “very severe”. As in our study, women ranked these functional disorders comparably to major surgical complications. In many cases, women are unaware of the risk of such outcomes, reflecting the need for better counselling patients about functional results before pelvic floor repair surgery. 

Comparison of the women’s and surgeons’ responses showed discordances in the vignettes describing the termination of a procedure, increased length of hospital stay, or complications leading to damage to physical integrity. Discordances in patient and physician perceptions of severity have already been reported in various diseases such as rheumatoid arthritis [14] and endometriosis [15]. However, data about surgical outcomes are scarce. Slankamenac et al. investigated perceptions of postoperative general surgery complications and showed that patients’ and physicians’ responses differed significantly [16]. More recently, Rendell et al. showed that patient-rated complication severity was discordant with provider-derived grading systems, and suggested the need to explore differences between patient and provider perspectives following surgery [12].

There is a need to improve our understanding of patient expectations and validate complication severity classifications. Other patient-centered decision scales have been developed for properties in women undergoing surgical treatment for PFD (the DRS-PFD, Decision Regret Scale-Pelvic Floor Disorders and SDS-PFD, Satisfaction with Decision Scales-Pelvic Floor Disorders) [17]. A few years later, Gutman et al. developed a Patient Centered Pelvic Floor Complication Scale (PFCS) which has been developed from the surgeons’ experience [18]. More recently, Fitzgerald et al. developed a simplified Pelvic Floor Complication Scale (simplified PFCS) which takes into account both surgeons and women’s point of view [19] and tested it in a multicenter cross-sectional study of patients and surgeons [20].

To the best of our knowledge, this is the largest survey published in the literature about patient perception of complication severity following surgery for pelvic floor disorders. We included women who had undergone surgery in different centers with various practices, which supports the representativity of our results. The clinical vignettes of our survey included the most frequently encountered situations reported in the VIGI-MESH registry, and were not restricted to postoperative complications. Questions were not about women’s own experiences; it is why we did not anticipate a recall bias.

Finally, the vignettes were presented in a random order and not according to the severity of the complication so as not to influence the answers.

Concerning POP repair, a higher proportion of women have had surgery with abdominal mesh placement. The surgical approach may have influenced the subjective perception of the operation, but in an abstract submitted to the International Continence Society congress and carried out from the VIGI-MESH registry, we found that the perceived quality of life of women and the subjective results were not different between the different surgical approaches [21].

Our findings may have been different if the women had been interviewed before their surgery. However, the study by Dunivan et al. reported that women in the preoperative groups shared similar perspectives regarding surgical expectations [13]. The vignettes were presented in the same random order for all women, and several women commented that they became more and more alarmed as they completed the questionnaire and stopped before the end of the survey. Finally, we did not assess the women’s literacy about surgical complications that may change their perception.

## 5. Conclusions

Women’s perception of complications following surgery for pelvic floor repair differs from the Clavien-Dindo classification. This classification does not take into account the repercussions of the complications in the real and daily life of the women. Thus, termination of the surgical procedure without repair, long-term symptoms of impairment, or recurrence after complication management are considered as serious complications by women.

There is a need to improve our understanding of patient expectations and develop and validate a complication severity classification that takes patient perception into account including the risk of long-term postoperative sequelae, the need to use medication or devices to help organ function (antibiotics, self-catheterization…), or the occurrence of another condition (urinary incontinence or pelvic pain). 

## Figures and Tables

**Table 1 jcm-11-03796-t001:** Respondents’ and non-respondents’ characteristics.

	**Non-Respondents** (*n* = 617)	**Respondents** **(*n* = 529)**	*p*
Age (years)	63.9 (11.6)	62.4 (11.7)	0.04
BMI (kg/m²)	25.6 (4.5)	25.2 (4.3)	0.23
Menopause			
No	149 (24.8)	153 (30.1)	0.05
Yes	451 (75.2)	355 (69.9)	
Type of surgery			
MUS alone	248 (40.8)	222 (42.7)	0.35
POP repair alone	284 (46.7)	246 (47.4)	
POP and MUS	76 (12.5)	51 (9.8)	
Complications			
No	566 (91.6)	473 (89.6)	0.25
Yes	52 (8.4)	55 (10.4)	

BMI: body mass index; MUS: mid urethral sling; POP: pelvic organ prolapse.

**Table 2 jcm-11-03796-t002:** Description and comparison of women’s and surgeons’ answers (percentages and Fisher’s exact test)—classifiable vignettes.

Clavien-Dindo Classification	Clinical Vignettes	Completion Order(Vignette #)	Women’s Answers (%) 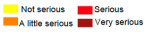	Surgeons’ Answers (%) 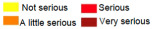	*p* **
I	Abdominal pain on D1—relieved by analgesics	11	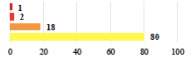	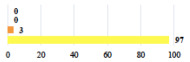	0.006
I	Urinary retention on D2 after POP surgery—placement of an indwelling catheter for a few days	13	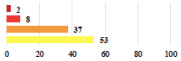	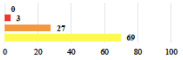	0.11
I	Bladder injury during MUS insertion—indwelling catheter for a few days	24	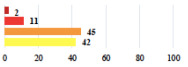	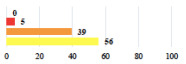	0.14
I *	Chronic urinary retention after MUS—self-catheterization	4	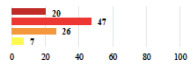	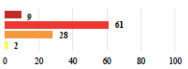	0.04
II *	Repetitive urinary tract infections 6 months after MUS—treated by antibiotics	9	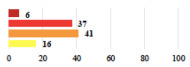	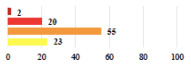	0.02
II	Laparoscopy scar re-opening—two stitches placed at the patient’s bedside under local anesthetic	3	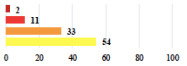	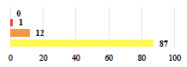	<0.0001
II	Bleeding during POP surgery—postoperative transfusion	22	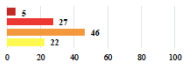	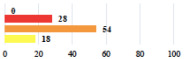	0.26
II *	Bleeding from the vaginal scar on D10 after POP surgery—hospitalization and vaginal compress placement	19	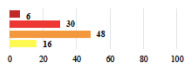	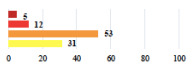	0.003
II	Pulmonary embolism after POP repair—anticoagulant treatment for 6 months	6	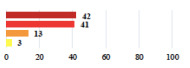	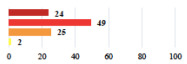	0.008
III	Severe pain after MUS—complete removal of the sling	1	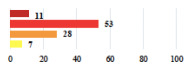	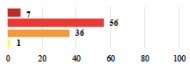	0.15
III	Vaginal erosion after MUS—partial removal of the sling	10	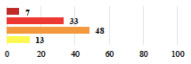	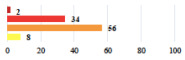	0.23
III	Vaginal erosion after MUS—partial removal with recurrence of SUI	28	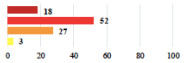	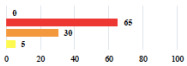	0.0002
III	Obstructive micturition on D1 after MUS—loosening of the sling	14	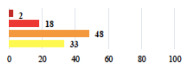	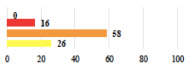	0.49
III	Severe urgency after MUS—bladder botulinum toxin injections	15	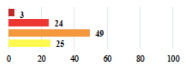	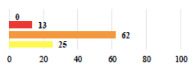	0.15
III	Vaginal evisceration on D15 after POP repair with hysterectomy—reoperation	5	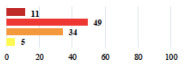	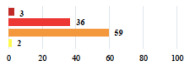	0.001
III	Ureteral obstruction on D3 after POP surgery—JJ ureteral stent for 3 months	12	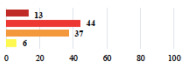	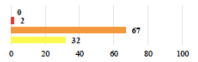	0.0005
III	Severe postoperative pain—release of sacrospinous fixation and subsequent POP recurrence	16	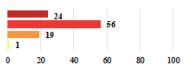	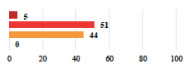	<0.0001
III	Severe dyspareunia—Transvaginal mesh removal	17	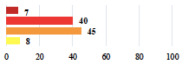	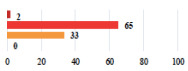	0.0006
III	Severe vaginal pain—Persistence of pain despite transvaginal mesh removal	20	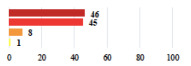	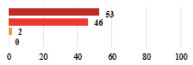	0.27
III	Bladder erosion of a transvaginal mesh—partial removal under cystoscopy	21	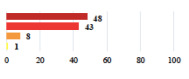	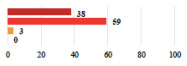	0.12
III	Hemoperitoneum on D1 after POP surgery—reoperation	26	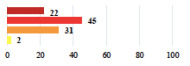	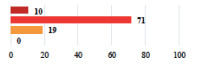	0.003
III	Vaginal erosion of laparoscopy placed meshes—removal of meshes	27	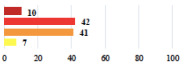	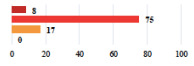	<0.0001
III	Rectal erosion of laparoscopy placed mesh—removal of mesh	29	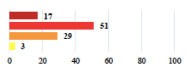	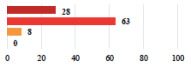	0.0009
IV	Peritonitis related to rectal injury on D1 after POP surgery—reoperation and 48 h of intensive care unit	18	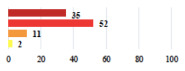	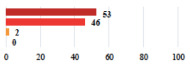	0.01
IV	Cardiac disorders following intraoperative hemorrhage—Cardiological intensive care unit	8	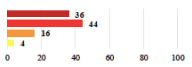	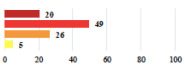	0.04

POP: Pelvic Organ Prolapse, MUS: mid-urethral sling, D1: day 1. * classified by analogy with another case described in Clavien-Dindo classification. ** Fisher’s exact test comparing women’s and surgeons’ responses.

**Table 3 jcm-11-03796-t003:** Description and comparison of women’s and surgeons’ answers (percentages and Fisher’s exact test)—unclassifiable vignettes.

Clinical Case	Completion Order	Women’s Answers (%) 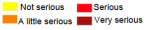	Surgeon’s Answers (%) 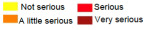	*p*
Conversion of laparoscopy to laparotomy during POP surgery	2	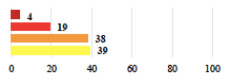	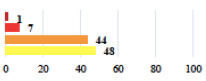	0.06
Vaginal hematoma after POP surgery—extension of hospitalization	23	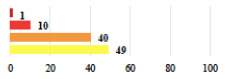	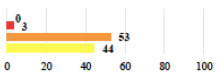	0.18
Abdominal adhesions—Laparoscopic procedure terminated without POP repair	7	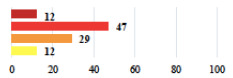	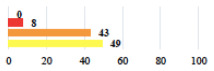	<0.0001
Rectal injury during laparoscopic sacrocolpopexy—procedure terminated without POP repair	30	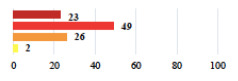	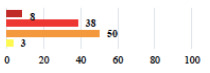	0.0006
Bladder injury during transvaginal mesh repair—vaginal repair without mesh placement	25	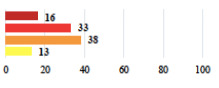	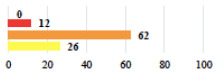	<0.0001

## Data Availability

Data are available from the corresponding author.

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
