# Peer review of "How Women Perceive Severity of Complications after Pelvic Floor Repair?"

_jcm, 2022, doi:10.3390/jcm11133796_

Round 1

Reviewer 1 Report

Interesting manuscript because it makes an investigation with an impact on the improvement of the Quality of Care and the Safety of Female Patients who have undergone surgery for Pelvic Floor problems by assessing the perception of the severity of surgical complications by the intervened women themselves and the surgeons, trying to overcome one of the weaknesses of the Clavien-Dindo Classification proposed by these authors in 1992 and modified in 2004, and which is currently the most widely used classification at international level to establish the complications of surgical interventions in 5 degrees.

It is therefore a timely clinical investigation considering the evaluation made by 529 women who have undergone pelvic floor surgery to assess the severity of the complications through 30 vignettes based on the complications and to estimate the severity by 4 degrees of None, Low Severity, Severe and Very Severe and qualitatively in the areas: Dependence or Limitation, Huillation or Distress, Disappointment or Failure, and Worry or Danger as outcomes compared with surgeons' results and ratings. Confluence or concordance was in 10 of the 30 vignettes.

Weaknesses:

The kappa concordance index of vignette-related observations between women and surgeons is not reflected. It is stated in the number of vignettes and percentage. Establishing concordance is timely among both women and surgeons.

It does not reflect the complications according to Clavien-Dindo that the women who participated in the study had, which may be a determinant or conditioning factor in their subsequent assessments.

The women's perception of severity differs from the classification made to evaluate surgical complications without taking into account the repercussions of the complications in the real and daily life of the women and, consequently, this is what the women who underwent surgery for pelvic floor problems express in their perceptions. In other words, women who have undergone surgery associate the seriousness of surgical complications with their repercussions on their lives, their well-being and their quality of life.

Reviewer 2 Report

Thank you for raising the relevant issue of discrepancies in patients' subjective perceptions of perioperative complications compared to the established systematic classification proposed by Clavien and Dindo. It could be a very good study. Some parts of the manuscript, e.g. the combination of tables and figures are attractive. In its present form, however, the submitted work suffers from too many fundamental flaws.

Major issues:

1) The significance of the differences was not assessed using statistical methods (although patient responses were quantified using a four-point Lickert scale).

2) The title and the entire manuscript do not state whether the pelvic floor reconstructions were performed vaginally or laparoscopically. Was the information about surgical approach (vaginal and laparoscopic) available? This differentiation would be of crucial importance since the surgical approach usually influences the subjective perception of the operation. If only vaginal surgeries were analyzed: This should be stated in the title, abstract and methods section. If not, the lack of differentiation should be considered a weakness of the study and addressed appropriately.

3) The abstract should be more informative. Please include more information into abstract, in particular the main categories in which the differences were significant (however, this presumes an adequate statistical evaluation).

4) Information on time-frame after surgery missing. This information is important because: a) Pelvic floor repair methods have evolved and b) a recall bias can contribute to the perception.

Minor issues:

1) If the authors point out the EUA validation of the Clavien-Dindo classification and point out the lack of classification of intraoperative events, the EUA classification of INTRAoperative events (published 2020) may offer the solution (PMID: 31787430). For a complete overview of classification systems of intra-, peri- and postoperative incidents in gynecology see PMID: 34691301.

2) Please remove the remnants of the mdpi template (first 3 lines of the "Results")

3) Only 15 references are included (1/ 3 of them related to general topics), while relevant publications in the field (subjective perception of POP surgeries, development of patient-centered scales for patients undergoing POP repair, etc.) are missing, PMID: 34608035, PMID: 30883438 , PMID: 27072172. Beyond these suggestions, conduct a thorough literature review and discuss your findings with at least 5-10 relevant studies.

Round 2

Reviewer 2 Report

I appreciate the authors' efforts to improve the paper. The manuscript has improved greatly. One information (from the author's answer) is missing from the manuscript ("we found that the perceived quality of life of women and the subjective results were not different between the different surgical approaches"). It is an important statement. Could you please integrate it into the paper? You can either cite the congress abstract or put "paper in preparation" in parentheses. This certainly does not minimize the publicability of the data.

Author Response

You are absolutely right.

The sentence below has been added at the end of the discussion:

“Concerning POP repair, a higher proportion of women have had surgery with abdominal mesh placement. The surgical approach may have influenced the subjective perception of the operation. But in an abstract submitted to the International Continence Society congress and carried out from the VIGI-MESH registry, we found that the perceived quality of life of women and the subjective results were not different between the different surgical approaches [21].”
